# Finite Element Analysis of Head–Neck Kinematics in Rear-End Impact Conditions with Headrest

**DOI:** 10.3390/bioengineering10091059

**Published:** 2023-09-08

**Authors:** Yuan Wang, Hanhui Jiang, Ee Chon Teo, Yaodong Gu

**Affiliations:** 1Faculty of Sports Science, Ningbo University, Ningbo 315211, China; jjhanhui@outlook.com (H.J.); guyaodong@nbu.edu.cn (Y.G.); 2Research Academy of Grand Health, Ningbo University, Ningbo 315211, China

**Keywords:** finite element, rear-end impact, cervical spine, whiplash, biomechanics

## Abstract

A detailed three-dimensional (3D) head–neck (C0–C7) finite element (FE) model was developed and used to dictate the motions of each cervical spinal segment under static physiological loadings of flexion and extension with a magnitude of 1.0 Nm and rear-end impacts. In this dynamic study, a rear-end impact pulse was applied to C7 to create accelerations of 4.5 G and 8.5 G. The predicted segmental motions and displacements of the head were in agreement with published results under physiological loads of 1.0 Nm. Under rear-end impact conditions, the effects of peak pulse acceleration and headrest angles on the kinematic responses of the head–neck complex showed rates of increase/decrease in the rotational motion of various cervical spinal segments that were different in the first 200 ms. The peak flexion rotation of all segments was lower than the combined ROM of flexion and extension. The peak extension rotation of all segments showed variation compared to the combined ROM of flexion and extension depending on G and the headrest angle. A higher acceleration of C7 increased the peak extension angle of lower levels, but the absolute increase was restricted by the distance between the head and the headrest. A change in the headrest angle from 45° to 30° resulted in a change in extension rotation at the lower C5–C6 segments to flexion rotation, which further justified the effectiveness of having distance between the head and the headrest. This study shows that the existing C0-C7 FE model is efficient at defining the gross reactions of the human cervical spine under both physiological static and simulated whiplash circumstances. The fast rate of changes in flexion and extension rotation of various segments may result in associated soft tissues and bony structures experiencing tolerances beyond their material characteristic limits. It is suggested that a proper location and angle of the headrest could effectively prevent the cervical spine from injury in traumatic vehicular accidents.

## 1. Introduction

During passenger vehicle collisions, which commonly occur on highways, whiplash is a common occurrence, whereby with head and neck, being the most flexible, exposed and unprotected segments of the human body, are the most vulnerable and frequent regions of injury in this traumatic, unexpected situation. Injuries to the cervical spine are dangerous as they are frequently linked to spinal cord damage, soft tissue injuries and hard bony structure fractures, which can be fatal or extremely damaging. In order to enhance prevention, diagnosis and therapy, it is crucial to understand the underlying mechanisms of this type of damage and dysfunction.

There is a dearth of useful data on whiplash injuries gleaned from in vivo investigations. Because of the participation of human volunteers, the harm threshold for the severity of impacts has not been successfully determined. Panjabi et al. [1,2,3,4] conducted a series of in vitro studies using whole cervical spine specimens to investigate potential injuries occurring during whiplash. The cervical specimen was positioned on a sled in a neutral position for their studies. Spring compression propelled an impactor, which then impacted the sled from behind. The sled, which was carrying the cervical spine specimen and the head surrogate, surged, achieved top speed and decelerated, at which point it broke and eventually came to rest. The motions of each motion segment and head were observed. The intervertebral rotations under different sled accelerations were also measured to predict the potential locations of neck injury. The head was represented by a steel model with a similar mass and moment of inertia to a typical human head in the sagittal plane. In addition, only the effects of peak sled acceleration were discussed.

FE models are commonly used in research to simulate changes in biological structures and characteristics that are difficult to measure, complementing experimental investigations [5,6]. The projected outcomes of the FE model also identify critical experiments that should be carried out [7]. Earlier FE models were separated into two fields: static study models and dynamic study models. The spinal geometries are often represented in more detail in the models created for static research [8,9,10,11]. Although these models may simulate strains, biomechanical responses and internal stresses under complicated loading situations, they cannot offer information about the whole spine column. Dynamic models are typically composed of a succession of vertebrae linked by ligaments and discs depicted as springs [12,13,14,15,16,17]. These models can only simulate the overall reaction to loads and are unable to represent the detailed interaction between the motion segments. 

Recent studies based on FE models of head and neck injuries have focused more on neck muscle activation [18,19,20]. Li Tiancheng et al. [18] built a complete FE model of a pilot’s head and neck. And they found out that during emergency ejection, the neck is protected by muscle pre-activation. In addition, Li Fan et al. [19] used FE models to study the head’s dynamic response to different frontal impact intensities in occupants in terms of muscle activation. According to the findings of these investigations, the combined dummy–human FE model provides good biofidelity for studying head and neck injuries. However, Li et al.’s study did not look into how the headrest’s angle and initial distance from the head during whiplash injury affected the head and neck [19]. The headrest plays an important role in preventing head and neck injuries and understanding the underlying mechanisms.

In recent research, a complete 3D FE model of the entire head–neck complex was constructed and utilized to predict the kinematic reactions of the head and cervical spine under both physiologic loading and rear-end impact. The influences of acceleration, distance between the head and headrest and headrest angles on the segmental motion of the cervical spine during whiplash were also discussed.

## 2. Materials and Methods

### 2.1. Modeling

In this study, a previously developed 3D FE model of the bony skull (C0) and cervical vertebrae (C1–C7), together with all associated articulated soft tissues, was used [21]. For the sake of completeness, here is a concise, in-depth explanation of the model development. Based on the embalmed C0 and C1–C7 vertebrae of a 68-year-old cadaver specimen, the coordinates of the surface profile of the bony structures (C0, C1–C7) were continuously captured with a flexible digitizer (FaroArm, Bronze Series, Faro Technologies, Inc., Lake Mary, FL, USA), and these data were later post-processed for the creation of an FE meshed model using ANSYS (ANSYS, Inc., Canonsburg, PA, USA). And the fundamental geometry for the intervertebral disc (IVD) modeling were derived from the average values given in the literature [22]. Each spinal motion segment, including cortical bone, cancellous bone, posterior elements, a disc annulus, a disc nucleus, an endplate and all ligamentous tissues associated with the motion segments between C0 and C7, were incorporated with their attachment points based on an anatomic text to generate the full C0–C7 model.

The skull was meshed with “quad” shell elements, whereas all of the vertebrae and discs were meshed with “brick” elements. To imitate the functioning of human ligaments, the majority of the ligaments were modeled using two-node nonlinear link elements, which only allow tensile axial force transfer. The C0–C7 superior and inferior facet joint articulations, the transverse ligament–odontoid process articulation and the dental–atlantal joint were all modeled using surface-to-surface interactions. To render the model stable for analysis, the articulating facet capsular ligaments and the alar ligament at the dental–atlantal joint were represented as “brick” components due to the unique C0–C1 articulation. Figure 1 depicts the finished C0–C7 FE model, which has 22,094 elements, 28,638 nodes and a global XYZ coordinate system. Figure 2 shows the C0–C7 FE model with a headrest.

The elements representing the bone head and bony spine components were considered to have linear elastic, homogeneous and isotropic material characteristics. Table 1 contains the specific spine material values that were obtained from the literature [9,11,23]. Furthermore, 8 “quad” shell elements with a certain thickness, Young’s modulus and density were meshed along the mid-sagittal of C0, so that the head is 5.5 kg in mass and has a moment of inertia of 0.035 kgm^2^, both of which are comparable to experimental specimens [4]. 

### 2.2. Validation

For validation, the C0–C7 FE model was tested in static flexion and extension circumstances using boundary and load configurations that were similar to those used in Panjabi et al.’s experimental research. [4] Therefore, a pure moment of 1.0 Nm was delivered sequentially on C0 along the sagittal anatomical plane to simulate the locomotions of the head and cervical spine in flexion and extension. All nodes on the inferior surface of the C7 vertebral body were entirely restrained. Then, the predicted kinematics characteristics, each motion segment’s range of motion (ROM) was examined, and the results were contrasted with the experimental data [4].

### 2.3. Whiplash Study

Using the same C0–C7 FE model, for the investigation of whiplash, the inferior surface of C7 was subjected to a 500 ms horizontal acceleration along the X-axis with a peak value of 4.5 G and a slower deceleration, as illustrated in Figure 3. During this process, the inferior surface of C7 was restrained from moving laterally or vertically. In addition, we introduced a square foam plate measuring 100 × 100 × 20 mm^3^ inclined at an angle 45° and 100 mm away from the head during the procedure. The foam plate’s rather large surface area ensured that the whole back of the head came into full contact with it. The headrest’s Young’s modulus was set at 2 MPa, which is comparable to that of soft foam material [24]. These circumstances were created to mimic the whiplash method used in previous studies [1,4]. After that, within the first 200 ms following impact, the expected responses of each motion segment were gathered and compared to the experimental data. Then, in an attempt to understand the role of input peak acceleration, distance between the head and headrest and headrest angle during a whiplash operation on the reaction of the head–neck complex, three extra analyses were carried out. For these additional analyses of peak acceleration/angle of headrest (i.e., 4.5 G/30°, 8.5 G/45° and 8.5 G/30°), appropriate changes were made in the FE model data file, and the computed results were investigated.

## 3. Results

### 3.1. Validation Analysis

Detailed comparisons between the projected overall main sagittal ROM (flexion and extension) for each locomotion segment and the results achieved by Panjabi et al. are shown in Figure 4 [4]. The model’s predicted ROMs and the experimental results were within one standard deviation. Because of the geometrical shape of the articulating joints and the expected nonlinearity of the ligaments, during flexion and extension, the whole structure’s projected moment–rotation relationships were extremely nonlinear. The C0–C1 section underwent the largest sagittal rotational motion of roughly 26° under either loading, followed by 20° in the C1–C2 section. The lower cervical segments between C2 and C7 experienced lower sagittal rotations between 8 and 10°, indicating that compared to the lower cervical vertebral level, the higher cervical segments (C0–C2 level) are significantly more flexible. Under flexion and extension, respectively, the rotation angles’ predicted ratios in terms of the C0–C2 section to the whole cervical spine were 47.3% and 55.6%. Additionally, the lower cervical levels (C3 to C7) showed few differences in ROMs. These results closely align with those found in experimental research [25,26]. The cervical spine’s normal motion in the sagittal plane under pure 1.0 Nm flexion and extension loadings was accurately approximated by the present model.

### 3.2. Whiplash Study

Figure 5 depicts the predicted rotational angle variation in every movement section during whiplash when the input peak acceleration was 4.5 G and the headrest angle was 45°. At different periods following impact, the flexion or extension angular rotational rotation peak varied among diverse segments. Except for C6–C7, all of the motion segments were in flexion within the first 10 ms following impact. Then, C3–C6 transitioned to distinct angulations of extension motion, whereas the upper segments (C0–C3) continued to move in flexion for a much longer period of time before switching to extension motion. Over a time period of 50–80 ms, the whole C0–C7 structure produced an S-shaped curve with bending at the top levels and extension at the bottom levels.

Over the 200 ms period, as shown in Figure 5, the C0–C1 and C1–C2 segments experienced flexion–extension–flexion motions, while the C6–C7 segments experienced extension–flexion–extension motions for different time spans.

Figure 6 shows the kinematic response of C0 with respect to C7 250 ms after impact. C0 accelerated immediately in posterior translation and extension rotation after impact until 110 ms and 130 ms, respectively, whereby it impacted the headrest and subsequently decelerated.

During the first 40 ms, C0 translated posteriorly by about 10 mm without any rotation or vertical motion, after which it continued to translate posteriorly and reached a maximum translation of 80 mm at about 110 ms, and the posterior translation of C0 decreased.

Coupled with posterior translation, the predicted results also show that C0 started to experience extension rotation from 40 ms onwards and reached a maximum extension angulation of 60° at about 130 ms, the time at which the rear region of C0 impacted the headrest; thereafter, the extension angulation reduced continuously. Figure 6 also shows there was a secondary minimal vertical translation of C0 with respect to C7 for the whole 220 ms analyzed after impact. The predicted results of translations and angular rotation of C0 with respect to C7 are different from experimental observations [2,3]; however, the fundamental alterations in the C0 movements and the intervertebral rotational angle followed comparable patterns and agreed with experimental findings [2,3].

Figure 7 and Figure 8 show the predicted peak rotational angle of flexion and extension in each motion segment during whiplash under different input conditions of G acceleration and headrest angle, respectively. Under any of these traumatic conditions, the predicted maximum flexion angular rotations of different motion segments, as shown in Figure 7, were lower than the combined flexion and extension rotational angles experienced at each corresponding motion segment under a normal static physiological loading of 1.0 Nm, as shown in Figure 4. However, the predicted extension rotations for each motion segment under all traumatic whiplash conditions analyzed were relatively greater than peak flexion rotations, and these values were also higher than those for the extension rotations experienced under static conditions. Under all the four traumatic conditions analyzed, C6–C7 consistently experienced much greater extension angles than when subjected combined flexion and extension motion under a physiological load of 1.0 Nm (Figure 4).

Under different G-acceleration inputs at C7 and headrest inclination conditions, the effect on the kinematics of the cervical spine was not explicit. For the peak flexion rotation of various segments, under different G accelerations with the same headrest inclination, there were fewer variations in peak flexion angles for most of the motion segments (C0–C5) compared to the many variations in the C5–C6 and C6–C7 segments (Figure 7). However, under different headrest inclinations, the variations in peak flexion rotations for different segments varied depending on the G acceleration. At 4.5 G with different headrest inclinations, the variations in peak flexion rotation were greater in the C2–C3, C5–C6 and C6–C7 segments. At 8.5 G with different headrest inclinations, greater variations in peak flexion rotations were observed in the C0–C1, C2–C3 and C6–C7 segments. No flexion rotation motion was observed in the C5–C6 segments under both 4.5 G and 8.5 G acceleration with the headrest inclined at 45°. This was not the case when the headrest was inclined at 30°, suggesting that the rear region of the translated and rotated C0 impacted the 30°-inclined headrest at a much earlier time, instead of at 130 ms, changing this segment’s extended motion to flexed motion.

The effects of headrest inclinations on C0–C7 segmental peak extension rotations are clearly shown in Figure 8. The segments between C0 and C5 showed greater variations in peak extension rotations then the C5–C7 segments in different headrest inclinations, irrespective of G acceleration. But under different G accelerations, only the C6–C7 segments showed significant variations in peak extension rotation.

## 4. Discussion

The aim of this research was to use a comprehensive 3D C0–C7 FE model to characterize the C0–C7 complex’s kinematics under both physiologic 1.0 Nm sagittal moments and a rear-end impact loading configuration. Based on geometrical information from a 68-year-old man’s skull and bony vertebrae that was collected using a flexible digitizer, and all associated ligamentous tissues from an anatomic text, an anatomically articulated C0–C7 FE model was generated. Validation of the FE model was conducted under flexion and extension load configurations, and the related anticipated outcomes of each motion segment were contrasted with the results of the experimental investigation [4]. The kinematic reaction of the entire head–neck complex in the anatomical sagittal plane under whiplash conditions was then investigated using the verified C0–C7 FE model.

In the validation research, the anticipated sagittal rotational angle for each motion segment was compared to the results found in the in vitro study [4], and the combined flexion and extension load configuration shows close agreement. The results show that compared to the lower cervical segments C3–C7, the upper cervical segments C0–C2 are substantially more flexible. The focus lies in the segmental kinematics at maximum load, rather than the absolute precise kinematics at each incremental load. These findings are in accordance with the experimental observations [24,27]. The motions of the C0–C1 and C1–C2 segments contribute to almost half (or higher) of the whole head–neck complex motion under sagittal moment loading conditions. This can be explained by considering the unique anatomically articulated characteristics of the C0–C1 and C1–C2 segments. There is no disc in the C0–C1 and C1–C2 levels; the articulation is just ligaments and joint articulations. Because of the lax ligaments in this area, complex, significant rotations are experienced under relatively small loads [3]. According to Panjabi et al.’s discussion in [4], a pure moment of 1.0 Nm may be established as the maximum physiological loading that will not cause head–neck complex damage. We believe that the C0–C7 FE model can accurately represent the motions of the human cervical spine under traumatic whiplash conditions; furthermore, according to this validation study, each motion segment’s associated maximum sagittal rotation angle can be used to predict if a head–neck injury from whiplash would occur.

The experimental result of this whiplash investigation was compared to the forecasted variations in rotation angles at various segmental levels [2,3]. The consequent angular deformation of the C1–C7 motion segments throughout the 50–80 ms period created an S-shaped curvature, which aligns with previous experimental research. The peak flexion angles of the cervical spine were all within their respective normal ranges 200 ms after rear-end impact. However, because of the abrupt, high acceleration of C7, the lower C6–C7 levels underwent hyperextension in the early period. This level might be a possible location of high risk during whiplash because the headrest has not yet been put into action and the peak extension angles for the lower cervical levels were reached during the early phase. Notably, the effects of muscular activation were not taken into account in the present study. According to Panjabi et al. [3], it takes relaxed muscle around 200 ms to build up enough force to restrain spinal motion. In our opinion, the validity of the prediction of head and neck movement in this study was not affected by the lack of muscle studies. Nevertheless, it should be pointed out that during the experiments, the specimens underwent a progressive deceleration with a pattern similar to that of an oscillation wave, which was slightly different from the deceleration employed in the present whiplash research. Furthermore, the preliminary distance between the 30°-inclined headrest and the head is not described in detail here. Therefore, the results from current study may not be exactly the same as the experimental data, but the basic motions of the cervical spine are properly reflected.

This whiplash study also showed that a higher acceleration of C7 increases the peak extension angle of the lower levels, which may increase the risk of injury at this location. However, the magnitudes of increases were not as significant as those obtained by Grauer et al. [2]. In their study, they reported the peak extension angle of C6–C7 at 8.5 G to be twice that at 4.5 G. This may be due to the relatively small initial distance between the head and headrest. In the current study, the initial distance between C0 and the 45°-inclined headrest was set as 100 mm. When the inclined headrest angle was at 30°, the distance between the head and headrest was reduced; as a result, the extended rotated head C0 impacted the headrest much quicker for 4.5 G and 8.5 G acceleration. The moment generated by the impact force blocked the head’s extension rotation, compelling the subjacent levels to prematurely revert to a state of flexion prior to attaining complete extension maturation. The impact of distance on the S-shaped kinematics during whiplash was examined by Garcia and Ravani [4]. They discovered that the cervical spine would not create an S-shape if there was less than 50 mm between the head and the headrest.

The headrest angle was also found to be an important factor for the motion of the cervical spine during whiplash. When the headrest was at a small angle, the head impacted the headrest much sooner and resulted in a lower extension angle for the lower levels. In the current study, all the upper levels (C0–C3) experienced lower extension angles when the headrest was at 30°. For C2–C3 in particular, the decrease in the peak value could reach 76% regardless of the input acceleration of C7. The restraining condition applied on C7 may have caused the absolute peak values of each motion segment to be inflated in comparison to the actual crash situation; however, the relative segmental motions estimated in this research should not be impacted. 

Furthermore, since the angle of the headrest can affect the impact force vector between the head and headrest, it may further influence the motions of the head–neck after impact. In this study, the exact point of impact between the head and the headrest, at either 30° or 45°, at different accelerations input was not investigated. We believe the impact force vector exerted on the head and the posture of the head/neck, creating a high-magnitude turning moment in any anatomical plane dynamically, may have a greater influence on the whole C0–C7 segmental rotation. Also, the changes in the motions of various segments from flexion to extension, or vice versa, occurring in a very short time, as shown in Figure 5, may result in associated soft tissue and bony structure injuries.

## 5. Conclusions

In summary, the C0–C7 FE model has clearly shown the magnitudes and rate of change in the segmental rotation and posterior translation of the articulated head and the cervical spine varied under different G values and headrest angles of inclination during whiplash in the 200 ms period. All segments, except C5–C6, flexed and extended under different Gs and different headrest angles of inclination. At 4.5 G and 8.5 G, with the headrest inclined at the angle of 45°, no flexion was observed for the C5–C6 segments. This was not the case for 30°, justifying the importance and effectiveness of having distance between the head and the headrest. The fast rate of changes in flexion and extension rotation of various segments may result in associated soft tissues and bony structures experiencing tolerances beyond their material characteristic limits. It is thus suggested that routine regular medical checkups be conducted for involved passengers.

## Figures and Tables

**Figure 1 bioengineering-10-01059-f001:**
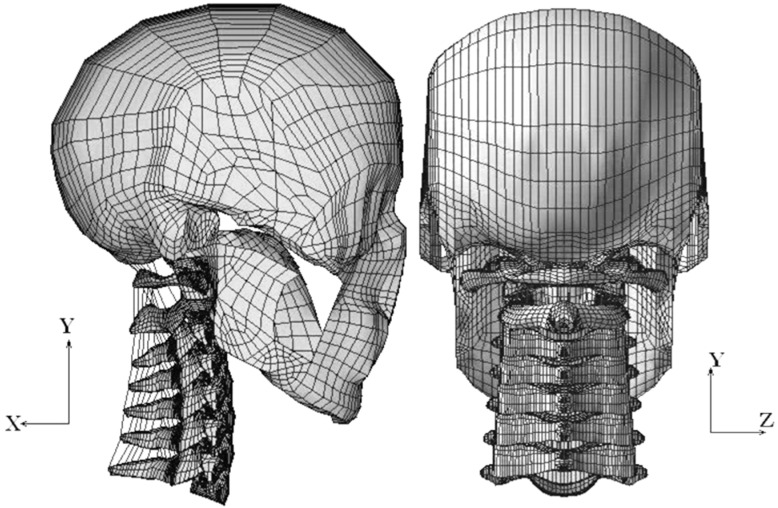
C0–C7 FE model (lateral and posterior views).

**Figure 2 bioengineering-10-01059-f002:**
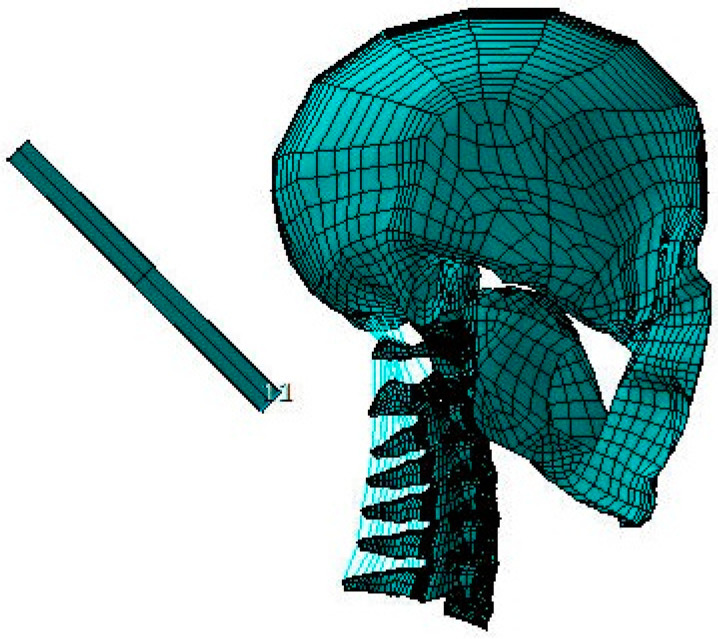
C0–C7 FE model with headrest.

**Figure 3 bioengineering-10-01059-f003:**
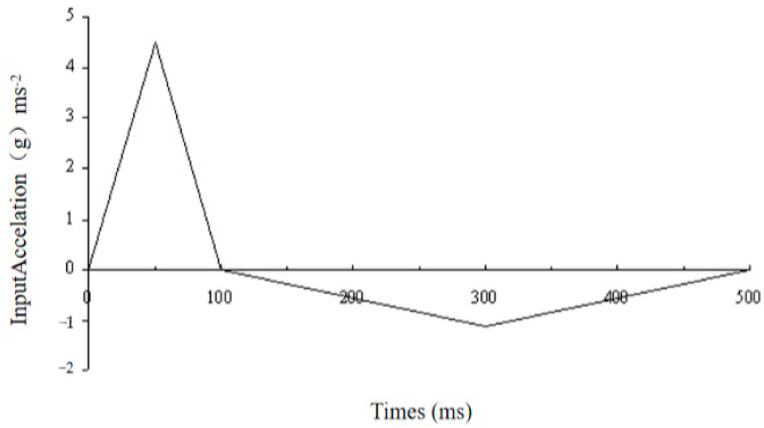
Input acceleration on C7 for dynamic simulation.

**Figure 4 bioengineering-10-01059-f004:**
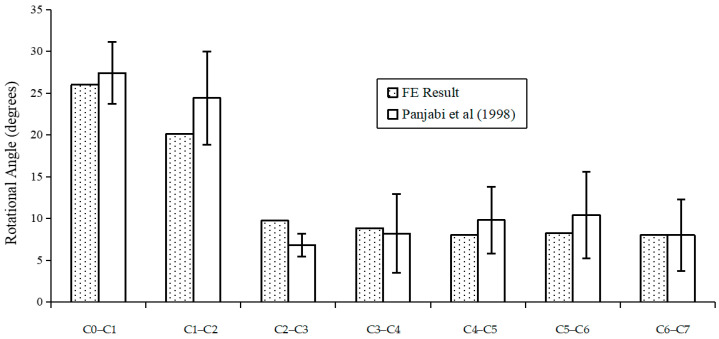
Comparison of ROM under combined flexion and extension with Panjabi et al. (1998) [1].

**Figure 5 bioengineering-10-01059-f005:**
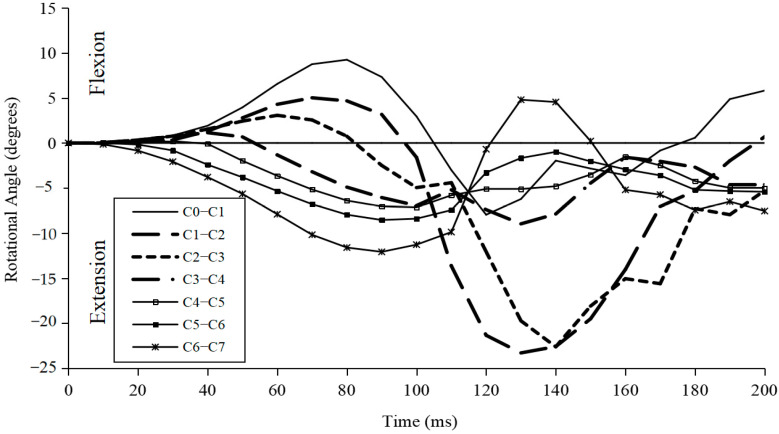
Predicted segmental motions during whiplash (4.5 G/45°).

**Figure 6 bioengineering-10-01059-f006:**
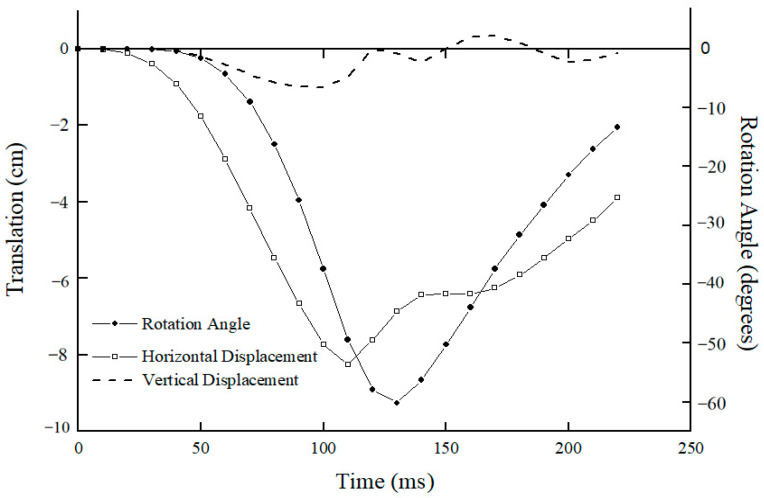
Predicted C0 motions with respect to C7 during whiplash (4.5 G/45°).

**Figure 7 bioengineering-10-01059-f007:**
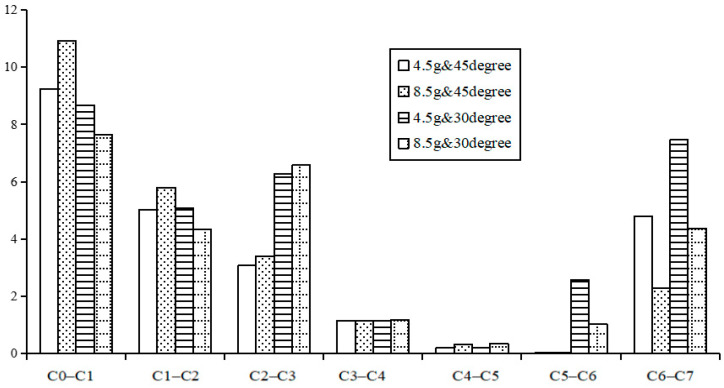
Predicted segmental flexion peak values during whiplash under different conditions.

**Figure 8 bioengineering-10-01059-f008:**
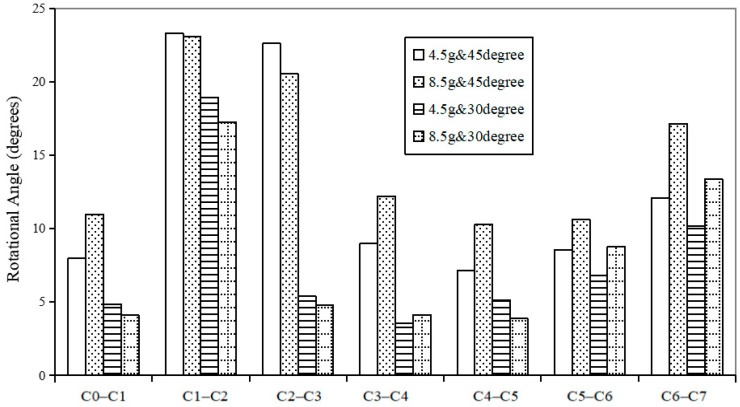
Predicted segmental extension peak values during whiplash under different conditions.

**Table 1 bioengineering-10-01059-t001:** Material properties used for various components in the current model.

Material	Young’s Modulus (MPa)	Poisson’s Ratio	Density (kg/mm^3^)
Cortical Bone	12,000.0	0.29	1.83 × 10^−6^
Cancellous Bone	450.0	0.29	1.00 × 10^−6^
Endplates	500.0	0.40	1.83 × 10^−6^
Posterior Element	3500.0	0.29	1.83 × 10^−6^
Annulus	3.4	0.40	1.20 × 10^−6^
Nucleus	1.0	0.49	1.36 × 10^−6^
Ligaments			
ALL	30.0	0.30	
PLL	20.0	0.30	
ISL, LF (C1–C2)	10.0	0.30	
SSL, ISL, LF (C2–C7)	1.5		
CL (C1–C3)	10.0	0.30	
CL (C3–C7)	20.0		
CL (C0–C1)	1.0		
AlL	5.0	0.30	
TL	20.0	0.30	
ApL	20.0	0.30	
Anterior Membrane	20.0	0.30	
Posterior Membrane	20.0	0.30	
NL	20.0	0.30	

ALL = anterior longitudinal ligament; PLL = posterior longitudinal ligament; SSL = supraspinous ligament; ISL = interspinous ligament; LF = ligamentum flavum; CL = capsular ligament; AlL = alar ligament; TL = transverse ligament; NL = nuchal ligament; ApL = apical ligament.

## Data Availability

The original contributions presented in this study are included in the article. Further inquiries can be directed to the corresponding authors.

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
