# Peer review of "Finite Element Analysis of Head–Neck Kinematics in Rear-End Impact Conditions with Headrest"

_bioengineering, 2023, doi:10.3390/bioengineering10091059_

Round 1
Reviewer 1 Report
The manuscript presents the development and validation of a three-dimensional head-neck finite element model that can simulate the responses of human cervical spine under static and dynamic conditions. This study discusses the effects of different factors, such as peak pulse acceleration, headrest angle, and distance between the head and the headrest, on the kinematic responses of the head-neck complex during whiplash. The authors conclude by suggesting that a proper location and angle of the headrest could prevent cervical spine injury during whiplash. Before it can be considered to be published, I recommend major revision with the following comments well addressed.
Line 101: The author said that “Most of the ligaments were modeled using two-node nonlinear link elements”. It should be clarified that which ligament was represented by the lines. How many link elements are there in the model?
Line 112: The author said that “some mass elements were added to some nodes…”. The author should specify how many mass element were added and which node has been attached with these mass element. Why not just assign the mass density of the whole head?
Line 114: What does kgm2 mean? If the 2 means square, the format should be revised.
Line 117: A figure for illustrating the boundary and load configurations should be added. How did the author apply the moment load i.e. 10Nm?
Line 127: The element type, material properties and the contact condition for the foam plate should be provided.
Line 143: I cannot find any figure for the moment-rotation relationships.
Figure 2, the label of vertical axis is incomplete.
A figure shows the deformation, and the stress distribution of the mesh model should be provided. Otherwise, it is difficult for the readers to understand the results of ROM variations.
Author Response
The manuscript presents the development and validation of a three-dimensional head-neck finite element model that can simulate the responses of human cervical spine under static and dynamic conditions. This study discusses the effects of different factors, such as peak pulse acceleration, headrest angle, and distance between the head and the headrest, on the kinematic responses of the head-neck complex during whiplash. The authors conclude by suggesting that a proper location and angle of the headrest could prevent cervical spine injury during whiplash. Before it can be considered to be published, I recommend major revision with the following comments well addressed.
Line 101: The author said that “Most of the ligaments were modeled using two-node nonlinear link elements”. It should be clarified that which ligament was represented by the lines. How many link elements are there in the model?
Answer: Thank you very much for your recommendation. All ligaments are represented by 2 noded link elements, and shows as lines in the model. There are 555 links elements (between c3-c7) active only in tension while 71 links elements between co-c3 are non linear elements with input data using load displacement curves.
Line 112: The author said that “some mass elements were added to some nodes…”. The author should specify how many mass element were added and which node has been attached with these mass element. Why not just assign the mass density of the whole head?
Answer: Thanks for your kind suggestion. We use 8 quad shell elements with thickness, density, E,etc to model the additional mass to make the head mass of 5kg and inertia value. I have supplemented it in the manuscript.
Line 114: What does kgm2 mean? If the 2 means square, the format should be revised.
Answer: Thank you for pointing out the error, we have made the correction in the manuscript.
Line 117: A figure for illustrating the boundary and load configurations should be added. How did the author apply the moment load i.e. 10Nm?
Answer: Thanks for your comments. We have revised the appropriate sentences to define how the load and boundary condition for this physiologic loading.
Line 127: The element type, material properties and the contact condition for the foam plate should be provided.
Answer: Thank you for your thoughtful suggestion. We have included the information of the foam plate and material property in the manuscript.
Line 143: I cannot find any figure for the moment-rotation relationships.
Figure 2, the label of vertical axis is incomplete.
A figure shows the deformation, and the stress distribution of the mesh model should be provided. Otherwise, it is difficult for the readers to understand the results of ROM variations.
Answer: Thanks for your comments. Figure 3 shows the rotation at various segments under combined 1.0Nm of flexion and extension moment applied at C0. Figure 2 is complete, vertical axis refers to acceleration in g. And no stress plots are provided as this paper mainly concerns about kinematic.
Reviewer 2 Report
• Overall impression:
The subject addressed in the paper is relevant for publication in Bioengineering in the scope of Biomechanics and Sports Medicine, although the way in which the information is presented is confusing and dispersed, becoming uninteresting for the reader. Particulary, it is unexpected to have a full section 3.3 with all figures and tables, disconected from the text in which are refered. Therefore, this paper must be improved in the negative issues, and the positive aspects are also highlighted in the following comments.
• Current references:
Most of the references are old, particularly about the mechanical properties of connection between vertebrae (elasticity), still the experimental procedure for obtaining these data is described in detail and with good illustrations in the works of Panjabi. It is suggested that in this submitted work the authors should make a greater effort in the descriptions of the models and respective figures.
• Research:
In the research, the authors distinguished between two distinct computational models, one for kinematic analysis (movement of the head and vertebrae) and another to verify the maximum angle of rotation of the vertebrae. This distinction is relevant since the developed model is to be used for both situations. The generated model is detailed so it may be useful in determining the location of the lesions along the spine in dynamic tests, given that with the current models, only the HIC and the neck injury criteria are calculated in the dynamic tests.
• Methods:
By scanning the bones of the skull and vertebrae, the authors were able to reproduce in detail the structure of the bones of the head and neck, however in the modeling of the discs were used data from the bibliography. It would have been more correct to try to evaluate in the same cadaver the disc dimensions because the bibliography values are different from the cadaver analyzed. In this part a question remains with the authors decision to use the bibliography data.
In the validation process, the authors were limited to verifying the magnitudes related to the static evaluation (ROM), however it would be more correct to perform two validations, one for the static and another for dynamics, applying the acceleration pulse at C7 in corpses (no muscle reaction) for displacements and rotations checking.
In the whiplash study the properties of the headrest foam are not defined nor is depicted an image of the setup of the dynamics tests. Also the 8.5 G pulse is not represented in figure 2, the question remains if the only difference between the 2 pulses is just the peak acceleration value or if there are more differences.
• Final Recommendations:
✓ Show the images of the experimental assembly with the headrest;
✓ Define the moment when the head contacts the headrest;
✓ Reorganize the information in the paper displaying the data close to where it is referenced in the text (if allowed) in particular graphics to facilitate consultation while reading.
Author Response
- Overall impression:
The subject addressed in the paper is relevant for publication in Bioengineering in the scope of Biomechanics and Sports Medicine, although the way in which the information is presented is confusing and dispersed, becoming uninteresting for the reader. Particulary, it is unexpected to have a full section 3.3 with all figures and tables, disconected from the text in which are refered. Therefore, this paper must be improved in the negative issues, and the positive aspects are also highlighted in the following comments.
- Current references:
Most of the references are old, particularly about the mechanical properties of connection between vertebrae (elasticity), still the experimental procedure for obtaining these data is described in detail and with good illustrations in the works of Panjabi. It is suggested that in this submitted work the authors should make a greater effort in the descriptions of the models and respective figures.
- Research:
In the research, the authors distinguished between two distinct computational models, one for kinematic analysis (movement of the head and vertebrae) and another to verify the maximum angle of rotation of the vertebrae. This distinction is relevant since the developed model is to be used for both situations. The generated model is detailed so it may be useful in determining the location of the lesions along the spine in dynamic tests, given that with the current models, only the HIC and the neck injury criteria are calculated in the dynamic tests.
- Methods:
By scanning the bones of the skull and vertebrae, the authors were able to reproduce in detail the structure of the bones of the head and neck, however in the modeling of the discs were used data from the bibliography. It would have been more correct to try to evaluate in the same cadaver the disc dimensions because the bibliography values are different from the cadaver analyzed. In this part a question remains with the authors decision to use the bibliography data.
Answer: At the early day when we do the modeling of this model, only bony skull and c1-c7 vertebrae were available, and for the disc we use data from anatomic text and model the whole head/neck complex with lordosis angle of about 37 degrees.
In the validation process, the authors were limited to verifying the magnitudes related to the static evaluation (ROM), however it would be more correct to perform two validations, one for the static and another for dynamics, applying the acceleration pulse at C7 in corpses (no muscle reaction) for displacements and rotations checking.
Answer: We did the validation based on literature data that we can compare against our FE results to validate the model and justify the use of it for further analysis.
In the whiplash study the properties of the headrest foam are not defined nor is depicted an image of the setup of the dynamics tests. Also the 8.5 G pulse is not represented in figure 2, the question remains if the only difference between the 2 pulses is just the peak acceleration value or if there are more differences.
Answer: We have amended the manuscript to in the headrest foam size and material. For 8.5G the peak value of input pulse changes from 4.5 to 8.5 while the time remains unchanged.
- Final Recommendations:
✓ Show the images of the experimental assembly with the headrest;
✓ Define the moment when the head contacts the headrest;
✓ Reorganize the information in the paper displaying the data close to where it is referenced in the text (if allowed) in particular graphics to facilitate consultation while reading.
Answer: Thank you for your recognition of the merits of this paper. We have improved the content of the manuscript by incorporating additional information to address your concerns as much as possible.
Reviewer 3 Report
Well done study. I would like to see some comments on the selection of the values for the g parameters (4.5 and 8.5) and headrest angle. It would also be good if there a figure showing the modelled spine under g loading.
Nothing major
Author Response
Well done study. I would like to see some comments on the selection of the values for the g parameters (4.5 and 8.5) and headrest angle. It would also be good if there a figure showing the modelled spine under g loading.
Answer:Thank you very much for your review and recognition of this manuscript. For the selection of the values for the g parameters (4.5 and 8.5) and headrest angle, as this is a simulation study, the authors are only trying to find the effect of g (4.5 and 8.5 g, approximately twice as much). And the authors believe that there is no need to display charts as the content already describes the loading and boundary conditions.
Round 2
Reviewer 1 Report
The authors addressed most of may comments. However, some references should be provided after "FE models are commonly used in research to simulate changes in biological structures and characteristics that are difficult to measure. complementing the experimental 55 investigations. "
For example:
Nishida, Norihiro, et al. "Posterior Fixation for Different Thoracic-Sacrum Alignments Containing a Thoracolumbar Vertebral Fracture: A Finite Element Analysis." World Neurosurgery (2023).
Author Response
Tanks for your kind comment. We have updated the new references.
Reviewer 2 Report
The authors must improve figure 3, dimensions are missing.
Some minor issues are detectei um texto editing.
Author Response
Thanks for your comment. We have improved the Y-axis of Figure 3.